# How the Digital Economy Empowers the Structural Upgrading of Cultural Industries—An Analysis Based on the Spatial Durbin Model

**Fengge Yao [1], Ying Song [1] and Xiaomei Wang [2,*]**

1 School of Finance, Harbin University of Commerce, Harbin 150028, China
2 School of Tourism and Culinary Arts, Harbin University of Commerce, Harbin 150028, China
* Correspondence: wangxiaomeihsd@sina.com

**Abstract:** The digital economy (De) is a major driving force in pushing structural improvements in the cultural industry. Theoretically, the De can empower the structural upgrading of the cultural industry by promoting technological innovation. Empirically, based on the provincial panel data of 31 regions in China from 2013 to 2020, this research utilizes the spatial Durbin model (SDM) to reveal the impact of the De on the structural upgrading of cultural industries. It also utilizes the mediation effect to test the path of the De on the structural upgrading of cultural industries through regional technological innovation. It is found that (1) the structural upgrading of the cultural industry shows significant spatial autocorrelation, and the eastern region is where the high–high cluster pattern predominates. (2) The De could successfully encourage the restructuring of the local cultural industry, and the spillover effect in space also promotes improving the organization of the cultural industry in neighboring regions. (3) A mechanism analysis shows that the De realizes the upgrading of cultural industry structure by enhancing regional technological innovation ability. Based on this, relevant policy recommendations are made to promote the upgrading of cultural industry structure in three dimensions: improving the level of the De, strengthening the ability of inter-regional cooperation, and promoting technological innovation. The study provides an important reference for the upgrading of China's cultural industry structure from the De's perspective.

**Keywords:** digital economy; cultural industry; upgrading of industrial structure; technological innovation; spatial Durbin model; spillover effect

## 1. Introduction

  The cultural industry is regarded as a representative of the sunrise industry and green industry because of its minimal resource usage, low pollution, high added value, etc. Its development has entered the fast lane and is gradually becoming a new engine of regional and national economic growth. To meet the new wave of technology and promote development, the cultural sector is significant, and it is necessary to accelerate the realization of the De to accelerate the growth of the cultural industry and seize the high point of cultural innovation and development. In terms of the stage characteristics of the growth of the cultural sector, the cultural industry has experienced the transition from a planned socialist economy to a capitalist economy, from the original benefit of the stimulation of the policy package [1], after the residents were satisfied at the level of material consumption to the current consumption upgrade [2], which focuses more on the demand for spiritual and cultural indulgence, which the scale and speed of the cultural industry have achieved. In particular, since entering the "new normal", the deep transformation of the economy and the rise of the technological revolution have had a profound impact on the cultural industry, and China's cultural consumption has been continuously upgraded, especially in how the residents' demand for high-level cultural consumption products is strong, but the cultural sector in China is facing the contradiction

that the output scale is growing while the output structure is insufficient to match. China's cultural industry is facing a contradiction between the growing scale of output and the lack of matching output structure, the problem of "structural excess" in China's cultural industry, and the serious problem of the ineffective supply of cultural products [3]. It is clear that China's culture sector is dealing with supply-side structural reform pressure; in other words, the cultural industry is about to undergo a transition and improvement.

From the viewpoint of kinetic transformation of the growth of cultural industry, under the new economic form of the De, information technology and the cultural sector have developed simultaneously, describing the traits of the digital economy and industrial digitization. Compared with the previous one, the factor inputs, the production mode, and the kinetic energy of industrial development have undergone qualitative changes. On the one hand, as a resource-intensive industry, cultural resources are the core resources that constitute the growth of the cultural sector, and the traditional cultural industry is subject to the characteristic of "non-storability", which reduces the degree of optimization of cultural resources. On the other hand, the cultural industry is innovation-intensive, and the level of product innovation can be enhanced through creative grafting. The De can not only contribute to reducing the development cost in the innovation of the cultural product creation mode, dissemination mode, and presentation form but also enhance the technical efficiency of the industry. The De can not only reduce the development cost but also increase the industry's technical effectiveness, thus realizing the transformation of cultural industry output from quantity to quality.

In transforming and upgrading the cultural industry, the De plays a significant part in facilitating the application of traction and data empowerment [4]. The De is focused on digital growth and completely unleashes the value of data elements by utilizing China's large data, vast market, and numerous application scenarios, drawing out data bits' potential, encouraging effective consumer market penetration throughout all facets of the cultural industry, and helping the cultural industry to raise quality and productivity.

The following sections of this paper are organized as follows: Section 2 is the literature review; Section 3 is the theoretical analysis and hypotheses; Section 4 is the research design and variable description; Section 5 is the empirical research; and Section 6 is the research findings and suggestions.

## 2. Literature Review

### 2.1. Digital Economy

The De refers to the application of digital information and communication technologies to traditional industries to enable companies to move away from geographical location dependence [5], transform business strategies, achieve digital transformation, and increase value production [6]. The De, as a new type of economic relationship, exists in all sectors and provides development power for the entire economic system [7]. On the one hand, digital payments can optimize traditional payment methods to be more convenient and timely. On the other hand, it can reduce production and transaction costs [8]. The content of research on the De can be summarized in two aspects: the measurement and impact of the De. There is no agreed-upon method for measuring De development yet; it is mainly assessed by machine learning algorithms [9], the multidimensional autonomous development index [10,11], and the edge computing method [12], where multidimensional creation of indexes is more common. The impact of the De is multifaceted; for example, the De helps to keep prices stable while minimizing inflation [13], maintains social governance mechanisms [14], promotes the green transformation of the region and adjacent regional economies [15], drives clean energy development [16], curbs carbon emissions [17,18], and achieve sustainable development [19,20]. By fostering green technological innovation and enhancing human capital, we can achieve high-quality urbanization and economic development [21]. Furthermore, the De has a good influence on the modernization of industrial structures, mainly through strengthening regional innovation [22] and fostering urban human capital and technological advancement [23].

While the De has many advantages, it also poses some potential risks. First is the data element. Huge databases can reduce search costs, but at the same time, it is important to prevent the monopolization of databases [24]. Second, there is the risk of unemployment due to cybercrime, information leakage, and automation [25]. Finally, the "spatially diverse" and "physical online platform" characteristics of the De have also created new regulatory issues [26,27], which require the strengthening of the legal system of the De [27], the establishment of a corresponding legal framework [28], and the improvement of national cybersecurity level [29].

## 2.2. Structural Upgrading of Cultural Industries

Over the past decades, cultural industries have received extensive attention from scholars as a driver of economic growth and local development [30–32]. While there is no uniform definition of cultural industry [33], some of the literature adopts cultural industry, some adopt cultural creative industry, creative industry, creative economy, etc. [30,33–36]. Finally, Gonzalez et al. confirmed that social networks have opened up the connection between cultural products and services from suppliers to clients, from which we can assume that the industry of culture and the artistic field associated with it are the same [37].

Research on cultural industries can be summarized from both macro and micro perspectives. From the macro level, cultural industries have a huge contribution to the national economy and even the global economy as a whole [1,4,38–41]. The development of artistic industries is also strongly associated with the state of the local economy and cultural resources, and the level of innovation efficiency of cultural industries is much greater in the central and eastern regions than in the central and western regions due to regional differences [42]. Cultural facilities in a region can contribute to the scale of cultural industry agglomeration and cultural industry structural upgrading by creating a cultural atmosphere, combining cultural and commercial activities, and promoting cultural productivity and consumption [43]. At the micro level, the profitability, size, type of ownership, and cash flow of cultural industry companies are all significantly positively correlated with capital structure, while dividend policy is negatively correlated [44]. Cultural industry firms show better development resilience in turbulent environments and adapt to economic downturns more than other industries [45], and firm development size is a key factor limiting investment efficiency [46].

Industrial structure upgrading is usually manifested in the rationalization of industrial structure as well as the heightening of industrial structure, while the heightening of industrial structure is built on the basis of rationalization [47,48]. This study intends to explore the perspective of heightened industrial structure from the perspective of the proportionality of the cultural industry sector and the enhancement of labor productivity. From the macro level, the De can increase the proportion coefficient of the cultural industry sector. According to the neoclassical economic growth theory, per capita capital stock and technological progress can promote the growth of per capita output. The De penetrates into the traditional cultural industry chain through digital technology, increases the per capita capital stock of the cultural industry sector, and realizes the capital deepening of the sector. This, in turn, promotes an increase in the sector's per capita output and the level of sectoral output. It further increases the proportion of the sector in the output value of the cultural industry sector, accelerating the evolution of the proportionality of the cultural industry sector. From the micro level, the De helps to improve the labor productivity of cultural enterprises. Endogenous growth theory points out that both knowledge spillover and "learning by doing" can improve labor productivity. On the one hand, knowledge has positive externalities, and cultural enterprises can realize independent innovation, liberate productivity, and increase labor productivity through the De platform. The success of some enterprises can provide a template for other enterprises to learn and emulate, which will ultimately lead to the improvement of labor productivity of the whole industry. On the other hand, giving full play to the effect of "learning by doing" can weaken the trend of diminishing marginal returns of capital through the accumulation of knowledge and better

enhance labor productivity. Thus, the De has promoted the evolution of the ratio coefficient of the cultural industry sector and the improvement of labor productivity. Therefore, this study focuses on measuring the heightened structure of the cultural industry from two aspects: the ratio coefficient of the cultural industry sector and labor productivity.

### 2.3. De and Structural Upgrading of Cultural Industries

The De's arrival directs the modernization and development of the sector of services [49]. The digital revolution influences the structural upgrading of the cultural industry not only from the consumer side but also permeates and changes the expression and market positioning of each production sector [50]. On the one hand, the De has developed a practical and effective online consumption network for customers thanks to its digital infrastructure and the quick advancement of related logistics technologies, increasing the efficiency of transactions and changing the consumption patterns of culture [3,51,52]. The upgrading of consumption drives the cultural industry's structure upgrading. On the other hand, by combining cutting-edge digital innovations in the sector, the De transforms traditional industries and becomes a new avenue for industrial expansion [53]. The rapid development of the De and the explosion of online platforms have removed language barriers and restrictions on the transport of commodities, while culture greatly impacts product and process innovation, increasing the industry's competitiveness of businesses [54], and traditional cultural manufacturers also enhance the governance of new business modules and enrich the types of cultural products through the digitization process [55,56].

In conclusion, the research on the De in promoting economic development, industry change, and industrial upgrading is quite rich, and it also provides an important logical reference for our research. However, these studies are mostly theoretical, along with a few empirical ones. This study is dedicated to revealing how the De has affected the structural modernization of the cultural industry from both theoretical and empirical aspects. It enriches and supplements the existing research results and, on this basis, provides suggestions and references for policy decision makers with regard to the research findings.

## 3. Theoretical Analysis and Hypothesis

### 3.1. De and Structural Upgrading of the Cultural Industry

In the "double-cycle" development pattern, high-quality economic growth puts forward higher requirements for the optimization of industrial structure. Traditional industries can ride on the "De", smooth the motion of production elements, improve the industrial digitalization system, and develop the consumer market [57]. Emerging cultural businesses, the digital derivation of traditional cultural products, the sharing economy and cost dynamic monitoring system, and a series of business links prompt cultural enterprises to upgrade and transform their business, business model, and management. The De, a new economic model relying on the Internet, is able to break geographical boundaries, make cultural products mobile, and rely on network sales to reduce storage costs and increase cash flow. This new model of cultural industry that breaks geographical boundaries also provides the possibility for the development of cultural activities in different regions and the synergistic development of the cultural industry. The first theory is suggested in light of the debate above.

**Hypothesis 1:** *The De can promote the upgrading of the structure of the cultural industry and form a spatial spillover effect.*

### 3.2. Mediating Effect of Technological Innovation

The De relies on the characteristics of digital technology such as "programmability" and "data homogenization" to reduce the cost of enterprise innovation activities, promote the flow of information within the enterprise, reduce the asymmetry of market information, break the time and space constraints, and enable people to participate in technological innovation activities at any time and any place. The De can promote the innovation ability

of enterprises, promote the technological upgrading of the traditional cultural industry, and have a significant positive impact on both the seniority and rationalization of industrial institutions and the use of the De to enhance entrepreneurial activity, prompt cultural enterprises to upgrade and transform their business operations, business models and management, improve labor productivity, and improve the competitiveness of the cultural industry. The second theory is suggested in light of the previous debate.

**Hypothesis 2:** *The De can realize the upgrading of cultural industry structure by improving the level of technological innovation.*

## 4. Model Construction and Variable Selection
### 4.1. Variable Description
#### 4.1.1. Explained Variables

Cultural industry structural upgrading ($Csh_{it}$). At present, it is common for academics to choose industrial structure-heightening indicators to quantitatively define the upgrading of industrial institutions. This paper draws on Chenery et al.'s approach [58] to obtain the cultural industry structural upgrading index by summing the product of the sectoral proportionality coefficient and sectoral labor productivity, as shown in Equation (1):

$$Csh_{it} = \sum_{j=1}^{n} \left( \frac{Y_j}{Y} \right) \left( \frac{LP_{jt} - LP_{jb}}{LP_{jf} - LP_{jb}} \right) \tag{1}$$

where $Csh_{it}$ denotes the structural upgrading index of cultural industry in year $t$ of region $i$, $Y_j$ denotes the income of the arts industry/library industry/mass cultural service industry/cultural market operation institution/relics industry, and $Y$ denotes the total income; $LP_{jt}$ denotes the labor productivity of the arts industry/library industry/mass cultural service industry/cultural market operation institution/relics industry (income of each sector/number of employees) and $LP_{jb}$ and $LP_{jf}$ denote the labor productivity at the beginning and completion of industrialization, respectively; and $j$ = 1, 2, 3, 4, and 5 denote the labor productivity of the arts industry and library industry, respectively. (Income/number of employees in each sector), $LP_{jb}$, and $LP_{jf}$ denote labor productivity at the beginning and completion of industrialization, respectively; $j$ = 1, 2, 3, 4, and 5 denote the arts industry, library industry, mass cultural services industry, cultural market operators, and cultural relics industry, respectively. According to Chenery et al.'s standard structural model, we convert the 1970-year dollar benchmark calculated in the original article into 2020 dollars with a conversion factor of 6.7 and then combine the average exchange rate of that year, and take the per capita income of 6469.8 yuan and 97,046.4 yuan as the starting and finishing points of industrialization, respectively.

#### 4.1.2. Core Explanatory Variables

Digital economy development index (De): This paper intends to measure the De development index from two levels: Internet development and digital financial inclusion. At the level of Internet development indicators, four secondary indicators are selected; namely, Internet penetration rate, number of Internet-related employees, Internet-related output, and number of mobile Internet users, and the number of Internet users per 100 people are used as the measurement standard for Internet penetration rate, the percentage of computer service and software employees is used as the measurement standard for the number of Internet-related employees, the total amount of telecommunication services per capita is selected as the measurement standard for Internet-related output, and the number of mobile phone users per 100 people is selected as the measurement standard for the number of mobile Internet users. The total number of telecommunication businesses per capita is selected as the measure of Internet-related output, and the number of cell phone subscribers per 100 people is selected as the measure of the number of mobile Internet users. At the level of digital financial development indicators, this paper draws on the measurement standard of China's digital inclusive finance index jointly compiled by the Digital Finance

Research Center of Peking University and Ant Gold Service Group and uses the entropy weight method to measure the digital economy development index of 31 provinces and autonomous regions in China, and the results of the measurement are shown in Table 1.

**Table 1.** De development level indicator system.

| Primary Indicators | Secondary Indicators | Tertiary Indicators | Indicator Attributes |
|---|---|---|---|
| De | Internet penetration rate | Number of Internet users per 100 people | + |
| | Number of Internet-related employees | Percentage of employees in computer services and software | + |
| | Internet-related output | Total telecom services per capita | + |
| | Number of mobile Internet users | Number of cell phone users per 100 people | + |
| | Digital Financial Inclusive Development | China Digital Inclusive Financial Index | + |

Note: "+" represents a positive effect of the indicator.

### 4.1.3. Mediator Variable

Technological innovation (Rd): The extensive use of big data can help enterprises better use advanced digital technology for embedded business analysis, more scientific and efficient algorithms as an analytical measurement tool to improve the accuracy of decision making, product innovation, optimization of business processes, breaking the information barriers, and improving quality and efficiency for the development of enterprises. Technological innovation has gradually become the power engine for enterprises to develop new business forms and new modes and realize digital transformation [54]. Therefore, this paper selects the level of technological innovation of enterprises as the intermediary variable and uses the number of inventions obtained by the selected enterprises in the same year as a representative.

### 4.1.4. Control Variables

In addition to the De, the structural upgrading of cultural industries is also influenced by other factors. With reference to the established research results, the following control variables are added to the model. First, the level of economic development (GDP) [42], expressed by GDP per capita; second, the industrial structure (Ind), expressed by the proportion of tertiary industry output in GDP; and third, government support (Gov), where government policy-guided cultural values can support cultural and creative industries and stimulate other new industries [59], and government financial support can motivate cultural creative companies to improve their technological innovation performance [39,60], tax policies and cluster support can promote the development of open innovation systems [61], and this manuscript measures government support in terms of the share of government fiscal expenditure in GDP; fourth, the level of foreign openness (Fdi), expressed in terms of foreign direct investment.

### 4.2. Model Construction

Because the level of structural upgrading of the cultural industry presents regional characteristics, it is also necessary to consider the spatial spillover effect when examining the effect of the De on the structural upgrading of the cultural industry. In spatial econometric modeling, comparing the SAR with the SEM, SDM is less prone to estimation bias and error. This manuscript decides to start the study with the help of the SDM and intends to examine the influence of the De on the structural upgrading of the cultural industry under the spatial weight matrix using the geographic distance matrix, and the spatial weight matrix is constructed as shown in Equation (2):

$$W = \begin{cases} 1/d_{ij}^2, \ i \neq j \\ 0, \ i = j \end{cases} \tag{2}$$

where $d_{ij}$ denotes the distance between the two places based on latitude and longitude calculations.

To circumvent the failure of parameter significance tests due to model heteroskedasticity and multicollinearity, this manuscript constructs a spatial econometric regression model after logarithmic treatment of each indicator as

$$
\begin{aligned}
Csh_{it} = \rho\sum_{i=1}^{n} w_{ij}Csh_{it} + \alpha_1 + \beta_1 De_{it} + \beta_2 Ind_{it} + \beta_3 Gov_{it} + \beta_4 Gdp_{it} + \beta_5 Fdi_{it} + \\
\gamma_1\sum_{i=1}^{n} w_{ij}De_{it} + \gamma_2\sum_{i=1}^{n} w_{ij}Ind_{it} + \gamma_3\sum_{i=1}^{n} w_{ij}Gov_{it} + \gamma_4\sum_{i=1}^{n} w_{ij}Gdp_{it} + \gamma_5\sum_{i=1}^{n} w_{ij}Fdi_{it} + \\
\mu_i + \delta_t + \varepsilon_{it}
\end{aligned}
\tag{3}
$$

In the above equation, $w_{ij}$ is the spatial weight matrix, $Csh_{it}$ is the structural upgrading indicator of cultural industry in province $i$ in year $t$, $\sum_{i=1}^{n} w_{ij}Csh_{it}$ is the spatial lag term of the explanatory variable $Csh$, De is the index of the core explanatory variable De in province $i$ in year $t$, and $Ind_{it}, Gov_{it}, Gdp_{it}, Fdi_{it}$ are the observed values of control variables affecting the structural upgrading of cultural industries in province $i$ in year $t$. $\sum_{i=1}^{n} w_{ij}De_{it}, \sum_{i=1}^{n} w_{ij}Ind_{it}, \sum_{i=1}^{n} w_{ij}Gov_{it}, \sum_{i=1}^{n} w_{ij}Gdp_{it}, \sum_{i=1}^{n} w_{ij}Fdi_{it}$ are the spatial lag term of the De as well as the control variables, $\rho$ is the spatial regression coefficient presenting the structural upgrading of cultural industries, $\alpha_1$ is a constant term, $\beta$ and $\gamma$ are K*1-dimensional parameter vectors, and $\mu_i$, $\delta_t$, and $\varepsilon_{it}$ are individual effects, time effects, and random perturbation terms, respectively.

### *4.3. Data Sources*

The panel data of 31 provincial administrative regions (except Hong Kong, Macao, and Taiwan) in China from 2013 to 2020 are selected as the samples in this paper. The data on cultural industry structure upgrading metrics are mainly from the China Statistical Yearbook of Culture and Cultural Relics and the China Statistical Yearbook of Culture and Related Industries. The data on the De development index system are from the China Statistical Yearbook, the Tertiary Industry Statistical Yearbook, the Information Industry Yearbook, the Digital Inclusive Finance Index of Peking University, and the National Bureau of Statistics. The data of control variable indexes are from the provincial, municipal, and autonomous regions' Statistical Yearbooks. For some missing data, this paper used the interpolation method to complete the data. The descriptive statistics of the variables are shown in Table 2.

**Table 2.** Descriptive statistics of the variable.

| Variable | Obs | Mean | Std.dev. | Min | Max |
|---|---|---|---|---|---|
| Csh | 248 | 1.876 | 1.496 | −1.321 | 6.678 |
| De | 248 | 0.423 | 0.152 | 0.196 | 0.982 |
| Rd | 248 | 8.190 | 1.579 | 3.466 | 11.166 |
| Gdp | 248 | 10.747 | 0.416 | 9.690 | 11.955 |
| Fin | 248 | 5.493 | 0.288 | 4.746 | 6.068 |
| Gov | 248 | 8.437 | 0.589 | 6.827 | 9.766 |
| Fdi | 248 | 11.308 | 1.508 | 7.179 | 14.825 |

## 5. Analysis of the Empirical Results

### *5.1. Spatial Autocorrelation Test*

#### 5.1.1. Global Spatial Autocorrelation Analysis

Before conducting the spatial econometric regression, it is necessary to conduct a preliminary analysis of the data set to examine whether it is suitable for the spatial econometric model. In view of this, this section uses the global Moran index to conduct a spatial autocorrelation test on the observed samples to verify whether there is spatial dependence in the structural upgrading of cultural industries.

The standard range of Moran's I index as a characteristic index reflecting spatial correlation is within the interval of [−1, 1]. When Moran's I index is greater than 0, the spatial correlation between neighboring regions shows a significant positive relationship. On the contrary, when Moran's I index is less than 0, the spatial correlation between neighboring regions shows a negative correlation. When Moran's I index is 0, then it represents that the spatial correlation between neighboring regions is randomly distributed.

As can be seen from Table 3, in 2014, the global Moran's index of regional cultural industry structural upgrading passed the significance test of 5%, while the rest of the years from 2013 to 2020 passed the significance test at the 1% level, and the value of Moran's I index is positive, which indicates that there is an obvious positive spatial correlation in the structural upgrading of the cultural industry. It verifies the rationality of using the spatial econometric model for the sample data of this paper.

**Table 3.** Moran's I index of cultural industry structural upgrading and De development level.

| Year | Csh | | De | |
|------|-----------|---------|-----------|---------|
| | **Moran's I** | ***p*-Value** | **Moran's I** | ***p*-Value** |
| 2013 | 0.233 *** | 0.004 | 0.211 *** | 0.004 |
| 2014 | 0.118 ** | 0.098 | 0.192 *** | 0.007 |
| 2015 | 0.241 *** | 0.003 | 0.168 ** | 0.015 |
| 2016 | 0.342 *** | 0.000 | 0.190 *** | 0.006 |
| 2017 | 0.211 *** | 0.007 | 0.161 ** | 0.019 |
| 2018 | 0.254 *** | 0.002 | 0.158 ** | 0.023 |
| 2019 | 0.354 *** | 0.000 | 0.170 ** | 0.015 |
| 2020 | 0.343 *** | 0.000 | 0.187 *** | 0.008 |

Note: "***, **" represent significance at 1% and 5% significance levels, respectively.

### 5.1.2. Local Spatial Autocorrelation Analysis

To further examine the regional structural upgrading of cultural industries and its temporal development, as well as the local spatial correlation characteristics, the local Moran's I scatter accumulation of the structural upgrading of cultural industries is analyzed with 2013 and 2020 as the time profiles, as shown in Figure 1.

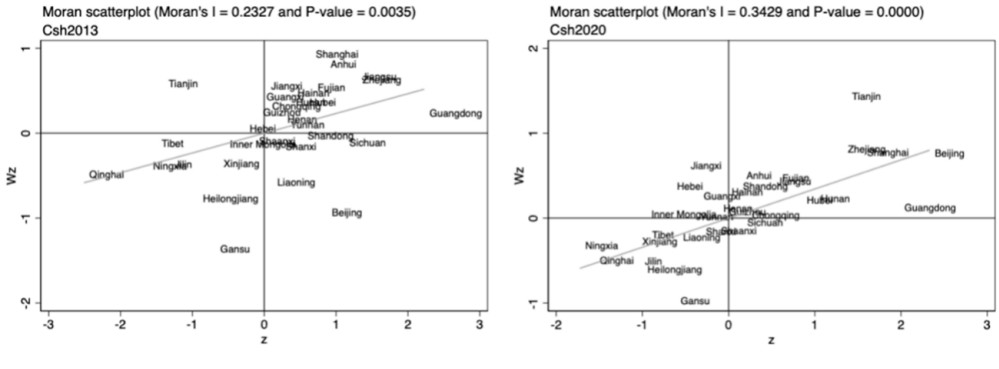

**Figure 1.** Csh local Moran's I index in 2013 and 2020.

From Moran's scatter diagram, we can find that more than 80% of the regions are located in the first quadrant of the "H-H" agglomeration and the third quadrant of the "L-L" agglomeration. Consistent with the global Moran index, the structural upgrading of the cultural industry as a whole shows a positive spatial correlation.

To present the changes in specific agglomeration areas more clearly and explicitly and to observe the spillover situation in different areas, the ArcGIS software was used to draw the regional cultural industry structure upgrading agglomeration map in 2013 and 2020, as shown in Figure 2.

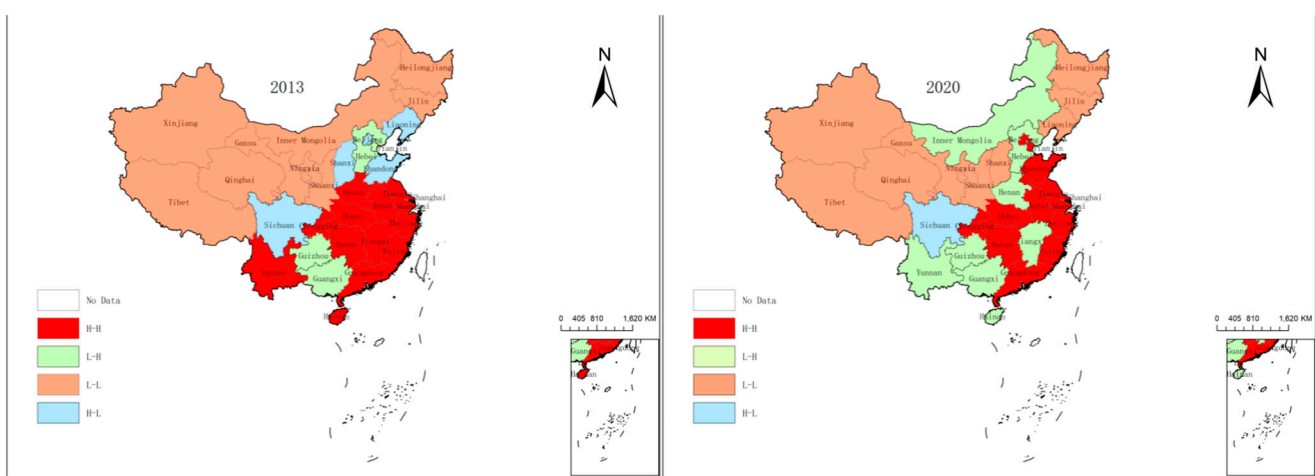

Note: The map is based on the standard map of the Map Technical Review Center of the Ministry of Natural Resources (review number GS(2020)4618), and the base map is unmodified.

**Figure 2.** Structural upgrading agglomeration map of cultural industry.

Specifically, most of the provinces and regions distributed in the first quadrant of the structural agglomeration map of cultural industries are located in the coastal Yangtze River Delta region, Fujian and Shandong Peninsula. The third quadrant is mainly in the regions of Tibet, Ningxia, Gansu, Qinghai, and Heilongjiang. The reason for the above imbalance in development may be the higher level of cultural productivity in developed regions and the relative abundance of factors such as capital, technological, and intellectual resources, as well as basic infrastructures. This is able to support the development of new types of cultural industries, such as creative design, animation and games, and network cultural services, etc., and such cultural products and services are more competitive in the market and have become an important part of the structure of the region's cultural industries.

*5.2. Model Selection*

Given that the spatial autocorrelation test has proven that the structural upgrading of cultural industries has significant spatial autocorrelation, a spatial econometric model can be constructed to explore the spatial spillover effect of the De on the structural upgrading of cultural industries. In order to select the optimal spatial panel model, a model selection test is required (see Table 4).

**Table 4.** Selection test of the spatial econometric model.

| Test | Statistical Values |
|---|---|
| LM spatial lag | 41.508 *** |
| Robust LM spatial lag | 13.007 *** |
| LM spatial error | 68.027 *** |
| Robust LM spatial error | 39.526 *** |
| LR_ Spatial lag | 18.190 *** |
| LR_ Spatial error | 23.780 *** |
| Wald _Spatial lag | 16.470 *** |
| Wald _Spatial error | 13.780 ** |
| Hausman test | 24.440 ** |

Note: "***, **" represent significance at 1% and 5% significance levels, respectively.

As can be seen from Table 4, the model passes the significance test of LM, which means the SDM could not be degraded into SAR and SEM. Based on this, the paper passed the Hausman test to compare the choice of using a random model or fixed model, and according to the test results, it passed the 5% significance level, so the fixed model was chosen. Further, the LR test was passed to determine the choice of time prescription, individual fixed or double fixed, and the test results all passed the 1% significance level,

indicating that the explanatory power of the two-way fixed effect was higher than that of other fixed effects. Therefore, this paper finally selects the SDM with the double fixed model for empirical study.

### 5.3. Benchmark Regression Analysis

According to the test results in Table 4, The SDM is chosen to investigate how the De affects structural improvement in the cultural industry and the spatial spillover effect. After the Hausman test, the fixed-effect SDM is selected, and individual, time, and double fixed are set; the estimation figures are displayed in Table 5. In model 1, the coefficient of spatial lag term ρ of cultural industry structural upgrading is significantly positive at a 1% significance level, indicating that there is a positive spatial spillover effect of China's cultural industry structural upgrading in space and that every 1% improvement in regional cultural industry structural upgrading will promote the improvement of the cultural industry structural upgrading in the neighboring regions by 0.267. This means that there is a significant spatial interaction effect among the provinces, and the cultural industry structural upgrading has a tendency of agglomeration in space and can promote the cultural industry structural upgrading of neighboring provinces through the diffusion effect. As a result, the research Hypothesis 1 was verified.

**Table 5.** SDM benchmark regression results.

| SDM | Csh | | | |
|---|---|---|---|---|
| | **Random Effect** | **Individual Fixed** | **Time Fixed** | **Two-Way Fixed Effect** |
| ρ(rho) | 0.560 *** | 0.529 *** | 0.516 *** | 0.267 *** |
| | (7.35) | (6.79) | (6.03) | (2.48) |
| De | 7.915 *** | 5.027 * | 10.738 *** | 6.351 ** |
| | (4.22) | (1.88) | (11.19) | (2.42) |
| Ind | −0.172 | −0.316 | 0.412 | −0.300 |
| | (−0.3) | (−0.57) | (0.53) | (−0.56) |
| Gov | 0.062 | 0.061 | 0.087 | 0.074 |
| | (0.54) | (0.56) | (0.51) | (0.7) |
| Gdp | −0.134 | −0.110 | −0.242 | 0.079 |
| | (−0.83) | (−0.68) | (−1.55) | (0.45) |
| Fdi | −0.054 | −0.040 | −0.215 *** | −0.051 |
| | (−1.04) | (−0.8) | (−3.07) | (−1.06) |
| W × De | −8.164 *** | −4.277 | 1.484 | 17.905 *** |
| | (2.77) | (−1.6) | (0.69) | (2.77) |
| W × Ind | −1.012 | −0.651 | −1.253 | 0.625 |
| | (−0.73) | (−0.49) | (−0.65) | (0.47) |
| W × Gov | 0.446 * | 0.436 * | 1.028 ** | 0.573 ** |
| | (1.62) | (1.68) | (2.50) | (2.23) |
| W × Gdp | −0.370 | −0.259 | −1.456 *** | 0.541 |
| | (−0.90) | (−0.63) | (−3.15) | (1.03) |
| W × Fdi | −2.000 | −0.219 | −0.341 * | −0.301 ** |
| | (−1.41) | (−1.64) | (−1.70) | (−2.24) |
| sigma2_e | 0.402 *** | 0.351 *** | 0.902 *** | 0.331 *** |
| | (10.25) | (10.92) | (10.82) | (10.98) |
| time | No | No | Yes | Yes |
| ind | No | Yes | No | Yes |

Note: "***, **, *" represent significance at 1%, 5%, and 10% significance levels, respectively.

### 5.4. Spatial Effect Decomposition

Due to the existence of the spatial lag term, the regression coefficients in the model no longer explain the actual impact of the explanatory variables on the variables being explained. Therefore, referring to the research idea of Wang et al. [62], the spatial effects of each variable are further decomposed by partial differential methods. As shown in Table 6, the estimated coefficients of the direct effect, indirect effect, and total effect of the De on the structural upgrade of the cultural industry all pass the significance test

at the 1% level, which indicates that every 1% increase in the De directly promotes the structural upgrade of the cultural industry in the region by 7.371% units, indirectly drives the structural upgrade of the cultural industry in the surrounding areas by 26.201% units, and the overall regional cultural industry structure upgrade by 33.572% units. Specifically, the direct effect of the De is significantly positive, demonstrating that the improvement of the local cultural industry structure will be positively impacted by the growth of the De. The specific reason may exist in having a perfect digital infrastructure in the era of the De, as the cornerstone of the combination of the De and cultural industry, regions can rely on the digital infrastructure and the application services it carries to promote the upgrading of cultural industry products and services so that there are many ways to increase the effectiveness of resource allocation, and thus promote the upgrading of the cultural industry structure.

**Table 6.** Spatial effect decomposition.

| Variable | Direct | Indirect | Total |
|----------|--------|----------|-------|
| De | 7.371 *** | 26.201 *** | 33.572 *** |
| | (2.78) | (3.28) | (4.14) |
| Ind | −0.295 | 0.725 | 0.43 |
| | (−0.56) | (0.41) | (0.22) |
| Gov | 0.111 | 0.775 ** | 0.887 ** |
| | (1.10) | (2.28) | (2.48) |
| Gdp | 0.107 | 0.794 | 0.902 |
| | (0.59) | (1.12) | (1.09) |
| Fdi | −0.068 | −0.409 ** | 0.477 ** |
| | (−1.45) | (−2.11) | (−2.22) |

Note: "***, **" represent significance at 1%, and 5% significance levels, respectively.

*5.5. Robustness Test*

In order to test the robustness of the baseline regression results, this paper uses the replacement of the spatial weight matrix and the lagged period to test the robustness of the baseline regression results separately. In this paper, the empirical results are tested by replacing the spatial distance inverse square matrix with the economic geography nested spatial weight matrix, and the estimation method and model remain the same as above. The specific results are shown in Table 7.

**Table 7.** Robustness test (I): replacing the economic–geographic nested spatial weight matrix.

| | Regression Results of the Replacement Space Weight Matrix | | Regression Results of the Original Model | |
|---|---|---|---|---|
| | Coefficient | Z-Statistic | Coefficient | Z-Statistic |
| De | 5.547 ** | 2.09 | 6.351 ** | 2.42 |
| W × De | 12.095 ** | 2.42 | 17.905 *** | 2.77 |
| Direct | 6.276 ** | 2.34 | 7.371 *** | 2.78 |
| Indirect | 16.886 *** | 3.08 | 26.201 *** | 3.28 |
| Total | 23.162 *** | 4.13 | 33.572 *** | 4.14 |

Note: "***, **" represent significance at 1% and 5% significance levels, respectively.

As can be seen from Table 7, the overall, direct, and indirect consequences of the De on the structural upgrading of the cultural industry all pass the significance level test of at least 5%. It indicates that between regions with close geographical and economic ties, on the one hand, the De, by penetrating and transforming various links of the value chain of the cultural industry, works both ways from both the supply and demand sides and realizes the cultural industry through the industry-linked sharing effect. On the other hand, the De has a significant network effect, with efficient information carrying and transmission, which can compress the space-time distance between regions and break geographical barriers

to a large extent. By promoting the cross-regional sharing of culture-related technologies, resources, and talents and constructing a new industrial supply chain in a complex network mode, it will promote the strengthening of the interaction and association of cultural industries between the local and neighboring regions and then effectively enhance the structural upgrading of the cultural industries in the local and neighboring regions. In the robustness test of replacing the weight matrix, the estimated results of the core explanatory variables are basically consistent with those obtained above, thus proving the reliability of the empirical findings.

In order to make the empirical results more robust, this paper lags the explanatory variables by one period, performs the robustness test again, and re-runs the regression, and the results are shown in Table 8. After lagging the cultural industry structure upgrading index by one period, the direct effect, indirect effect, and total effect of De on cultural industry structure upgrading are all positive and significant at a 1% level. This suggests that the De has a favorable impact on local and surrounding regions. This is consistent with the results of the previous paper and again verifies the robustness of the empirical results of this paper.

**Table 8.** Robustness test (II): variables lagged by one period.

| | Regression Results with One Period Lag | | Regression Results of the Original Model | |
|---|---|---|---|---|
| | **Coefficient** | **Z-Statistic** | **Coefficient** | **Z-Statistic** |
| De | 0.225 *** | 2.95 | 6.351 ** | 2.42 |
| W × De | 0.485 *** | 3.26 | 17.905 *** | 2.77 |
| Direct | 0.265 *** | 3.39 | 7.371 *** | 2.78 |
| Indirect | 0.845 *** | 4.11 | 26.201 *** | 3.28 |
| Total | 1.110 *** | 4.98 | 33.572 *** | 4.14 |

Note: "***, **" represent significance at 1% and 5% significance levels, respectively.

### 5.6. Mechanism Test

The core driving force for structural upgrading of the cultural industry is technological innovation, and the De can empower traditional culture and digitally innovate the cultural industry in terms of presentation, means of communication, and consumption methods. Traditional culture is creatively transformed using new creativity and design to achieve creative transformation and innovative development. Therefore, the higher the level of development of the De, the more sustainable the cultural and creative products and projects can be, which in turn promotes the structural upgrading of the cultural industry.

To this end, the number of inventions obtained in each region in the current year is selected and taken as a logarithm (Rd), and a mediating effect model considering spatial effects is constructed to test whether the De promotes the structural upgrading of the cultural industry by influencing technological innovation. The specific model setting form is as follows:

$$Csh_{it} = \rho_1 \sum_{i=1}^{n} w_{ij} Csh_{it} + \alpha_1 + \beta_1 De_{it} + \gamma_1 \sum_{i=1}^{n} w_{ij} De_{it} + \beta_1 Control_{it} + \gamma_1 \sum_{i=1}^{n} w_{ij} Control_{it} + u_i + \delta_t + \varepsilon_{it} \tag{4}$$

$$Rd_{it} = \rho_2 \sum_{i=1}^{n} w_{ij} Rd_{it} + \alpha_2 + \beta_2 De_{it} + \gamma_2 \sum_{i=1}^{n} w_{ij} De_{it} + \beta_2 Control_{it} + \gamma_2 \sum_{i=1}^{n} w_{ij} Control_{it} + u_i + \delta_t + \zeta_{it} \tag{5}$$

$$Csh_{it} = \rho_3 \sum_{i=1}^{n} w_{ij} Csh_{it} + \alpha_3 + \beta_3 De_{it} + \gamma_3 \sum_{i=1}^{n} w_{ij} De_{it} + \theta M_{it} + \tau \sum_{i=1}^{n} w_{ij} Rd_{it} + \beta_3 Control_{it} + \gamma_3 \sum_{i=1}^{n} w_{ij} Control_{it} + u_i + \delta_t + \varepsilon_{it} \tag{6}$$

Among them, model (4) tests the spatial effect of the De on the structural upgrading of cultural industries, model (5) tests the spatial effect of the De on technological innovation, and model (6) adds technological innovation and the De together to the explanatory

variables to test their effect on the structural upgrading of cultural industries. Model (4), model (5), and model (6) together form a mediating effect test model considering spatial factors. The final test results are shown in Table 9.

**Table 9.** Mediation effect test.

|  | Variables | Csh | Rd | Csh |
|---|---|---|---|---|
| X item | De | 6.351 ** (2.42) | 1.996 *** (3.17) | 6.125 ** (2.24) |
|  | Rd |  |  | 0.667 ** (2.52) |
| W × X item | W × De | 17.905 *** (2.77) | −4.874 *** (−3.17) | 18.094 *** (2.63) |
|  | W × Rd |  |  | 1.119 (1.45) |
| Direct | De | 7.371 *** (2.78) | 2.110 *** (3.28) | 7.108 *** (2.59) |
|  | Rd |  |  | 0.714 *** (2.7) |
| Indirect | De | 26.201 *** (3.28) | −4.676 *** (−3.2) | 25.611 *** (2.85) |
|  | Rd |  |  | 1.670 * (1.69) |
| Total | De | 33.572 *** (4.14) | −2.566 * (−1.94) | 32.719 *** (3.64) |
|  | Rd |  |  | 2.384 ** (2.19) |

Note: "***, **, *" represent significance at 1%, 5%, and 10% significance levels, respectively.

The regression results in Table 9 show that the De has a significant boosting effect in enhancing the technological innovation capacity of both local and neighboring regions. When technological innovation and the De are put into the same model for testing, it is found that the De continues to have a large enhancing impact on the structural upgrading of cultural industries in both local and neighboring regions. Technological innovation can significantly contribute to the upgrading of the structure of cultural industries in the region but not in the neighboring areas. In addition, after adding the variable of technological innovation, the coefficient of the impact of the De on the structural upgrading of cultural industries shows a certain decrease, indicating that technological innovation plays a part in mediating the effect of the role of the De on the structural upgrading of the cultural industry.

## 6. Conclusions and Recommendations

### 6.1. Conclusions

This paper constructs a De index system, measuring the level of upgrading the structure of the regional cultural industry, taking technological innovation as the mediating variable, and using SDM to verify the impact of the De on the upgrading of the cultural industry structure and the spatial spillover effect. The results show that, firstly, the De and the upgrading of the cultural industry structure are significantly and positively correlated in space, and the "high-high" agglomeration state is mainly concentrated in the eastern region, while the "low-low" agglomeration is mainly distributed in the western region. Secondly, after a series of model selection tests, the analysis of SDM concludes that the De can promote the structural upgrading of the cultural industry, and it has a spatial spillover effect and passes a series of robustness tests. The final mechanism analysis concludes that the De can realize the structural upgrading of the cultural industry by promoting technological innovation.

*6.2. Recommendations*

Based on the above conclusions, the following recommendations are made.

First, the transformation and upgrading of the cultural industry should borrow strength from the De. In terms of business operation, it should promote the digital derivation and extension of traditional cultural products with the help of digital operation; in terms of business model, it should transform the new development concept and use digital production factors to drive the sharing economy model to graft with the cultural industry; in terms of management, it should promote the digitization of management systems and monitoring systems to improve operational efficiency and reduce regulatory costs. Through the transformation and upgrading of the traditional cultural service industry, we will focus our attention on improving the quality and efficiency of the cultural service industry and continuously improve the production efficiency and output value of the cultural service industry.

Secondly, it is necessary to strengthen the digital technology cooperation between regions and the linkage development of cultural industries. Full play should be given to the linkage marketing ability of digital information technology, and through the "multi-network fusion" of cultural services in different regions, cultural resources should be aggregated to realize the precise matching of supply and demand of cultural projects and products and realize data realization. At the same time, the mechanism of inter-regional synergistic development is utilized to make up for the short boards of the De development in backward regions, giving serious consideration to the De's ripple effects and thus realizing the structural transformation of local and cultural industries.

Thirdly, while continuously improving the production efficiency of the traditional cultural industry through innovation, the new generation of the De and network technology is fully utilized to cultivate new forms of cultural industry, continuously creating new profit models for the cultural industry so as to continuously improve the efficiency and quality of industrial development, cultivate new comparative advantages of cultural industry, and contribute to the structural upgrading of cultural industry.

*6.3. Limitations*

This study has some limitations. First, due to data availability, the construction of this indicator of the De is based on existing research, and other attempts can be made in the future. Second, the mediating effect analysis only considered technological innovation, and future research can supplement other variables. Finally, regarding the data of government support in the control variables, it is better to use the data of government expenditure on cultural industry. At present, due to the lack of data, the government's general expenditure is used as a substitute for the time being, so the subsequent research can be improved here.

The main innovations of this study are as follows: first, the innovation of research perspective. Most current literary works on the De concentrate on the research of the De on enterprises, cities, and other micro and meso levels and seldom involve the upgrading of the structure of the cultural industry at the macro level. Therefore, it is innovative for this study to locate the research perspective at the macro level of cultural industry structure upgrading. Second, the research content is innovative. At present, the evaluation of the De's effects mostly centers on economic growth, industrial productivity, service trade, industrial structure, etc. This paper shifts the research content to the effect of the De on the modernization of the cultural industry's structural elements, which is an innovation of the research content in this field. Third, the research method is innovative. This study uses SDM to assess the effects of regional spillover and spatial correlation between the De and cultural industry structure upgrading and passes a series of robustness tests. This is a complementary and innovative approach to the related research topic using empirical analysis methods.

**Author Contributions:** All authors contributed to the study's conception and design. Material preparation, data collection, and analysis were performed by Y.S. The first draft of the manuscript was written by Y.S. X.W. provided mentoring support, F.Y. provided research funding, and all authors commented on previous versions of the manuscript. All authors have read and agreed to the published version of the manuscript.

**Funding:** This research received no external funding.

**Institutional Review Board Statement:** Not applicable.

**Informed Consent Statement:** Not applicable.

**Data Availability Statement:** All data generated and analyzed during this study are included in this manuscript.

**Conflicts of Interest:** The authors declare no conflict of interest.

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
