# Peer review of "How the Digital Economy Empowers the Structural Upgrading of Cultural Industries—An Analysis Based on the Spatial Durbin Model"

_sustainability, doi:10.3390/su151914613_

Round 1

Reviewer 1 Report

Dear authors,

First of all, I would like to express my thanks and congratulations for the study conducted.

I kindly request you to seek proofreading for the English language as the expression is not always clear.

Additionally, I urge you to revise the abstract in terms of clarity, including by shortening sentences to enhance clarity.

In the conclusion section, please present the connection between your obtained results and your recommendations in a clear form.

Please highlight:

-       Your contribution to advancing knowledge in the field,

-       Who can benefit from your study,

-       Why or to what extent your results, obtained based on data from 31 provinces of China, are relevant for enriching knowledge in general.

Good luck!

Reviewer 2 Report

1. What is the main question addressed by the research?  

The Research is focusing in Digital Economy applying the Durbin Model and their effect in the cultural industry in 31 regions in China from 2013 to 2020.

2. Do you consider the topic original or relevant in the field? Does it
address a specific gap in the field?  

From my point of view is a original due to apply the Durbin Model in the Digital Economy.

3. What does it add to the subject area compared with other published
material?

Probably is quiete new since try to apply the new era in de Digital economy.

4. What specific improvements should the authors consider regarding the
methodology?

What further controls should be considered?   The spatial econometric regression after the application of logarithmic treatment.  

5. Are the conclusions consistent with the evidence and arguments presented and do they address the main question posed?  

Yes, but probalby it needs more details about the conclusions  

6. Are the references appropriate?  

Yes, the references are appropiate and actually since the digital economy is a big challenge.  

Reviewer 3 Report

Title: “How the Digital Economy Empowers the Structural Upgrading of Cultural Industries-An Analysis Based on the Spatial Durbin Model”

The article is prepared on a current topic; the article is of interest to scientists; and the article can be published, but it needs preliminary thorough revision. The article is devoted to the analysis of the digital economy impact on the structural upgrading of cultural industries.

Notes and recommendations for authors:

1) in my opinion, it is worth specifying the research question at the beginning of the article

2) I think, it is advisable to clearly identify the hypotheses of the study; and in the section Conclusions to determine how these hypotheses were proven within the study;

3) the section Discussion should be developed (with the supplementing it with links to specific studies and publications by scientists who have studied similar research problems);

4) at the end of the article, it is worth indicating the limitations of the study;

5) Abstract should indicate the purpose of the article;

6) should expand the list of keywords;

7) reference should be made in accordance with the requirements of the journal 

Reviewer 4 Report

Based on the digital economy development and cultural industry data of 31 regions in China from 2013 to 2020, this manuscript applies the spatial Durbin Model to examine the influence mechanism and spatial spillover effect of the digital economy on the structural upgrading of the cultural industry and also introduces innovation as a mediating variable to explore the mechanism of the role of the digital economy in promoting the structural upgrading of cultural industry. Overall, I think this research topic is meaningful and the manuscript has a certain workload. A few suggestion for your reference. 

# 1. The problem awareness of the article is insufficient, and scientific problems are not clearly raised in the Introduction and Literature Review section.

# 2. In III. Model construction and variable selection, the selection of control variables lacks relevant theoretical and literature support, and some key control variables are not selected. A more obvious problem is that the article measures government support by the proportion of government fiscal expenditure to GDP (see Lines 217-218). However, there are a large number of government fiscal expenditure items, and the author should select the expenditure items related to the cultural industry.

# 3. Finally, I found that there are many typos and sentences with incorrect English grammar. Please recheck English with a professional native speaker.

Extensive editing of English language required

Reviewer 5 Report

In the introduction and review part of the article, the authors present a number of statements that require support from the results of earlier studies. For example: "The De can not only play a role in reducing the development cost in the innovation of cultural product creation mode, dissemination mode, and presentation form but also enhance the technical efficiency of the industry, thus the De can not only reduce the development cost but also improve the technical efficiency of the industry, thus realizing the transformation of cultural industry output from quantity to quality.” OR "The rapid development of the De and the rise of digital platforms have removed language restrictions and barriers to the distribution of goods".   A review of the literature on the topic of the digital economy does not provide a complete picture of it. The authors write only about the positive effects of digitalization, but there are also negative ones. It should also be taken into account that moving away from a geographic location is typical for truly digital companies. While for many companies, incl. digitally active ones, place still matters.   The object of the study is Cultural industries. The authors need to clarify what they mean by this concept. It is necessary to indicate the set of industries under study.   The composition of the indicators included in the Digital Economy Development Index requires additional justification. Some indicators (for example, Number of Internet users per 100 people; Number of cell phone users per 100 people) reflect the conditions for the development of the digital economy, but not the digitalization of companies themselves. I would advise the authors to enrich the analysis with literature on the digital maturity of companies.   The inclusion of the China Digital Inclusive Financial Index in the system of indicators also requires justification. The composition of the Index is not specified.   It is advisable for the authors to indicate full links to data sources in the "Data sources" section, in order to be able to verify the results.   Exploring spatial patterns, it would be good for the authors to give a geographical map with the results obtained. And also add a description of the identified spatial dependence between the regions of China.   At the end of the article, you need to add a section with the limitations of the study. It is also necessary to better reflect the significance of the results obtained by the authors. How do the results of the authors compare with the results of other studies, for example, those that revealed the phenomenon of the virtual agglomeration of creative industries in China?

Round 2

Reviewer 4 Report

The author has made careful revisions to the manuscript, which can be accepted in the present form.

Author Response

Thank you again for your valuable comments that led us to this better version.

Reviewer 5 Report

The authors have carried out a significant amount of work on the text of the manuscript. However, the article can still be improved. First of all, further revision is required on the literature review. In its current form, it is too general, especially in terms of describing the digital economy. A clearer connection to the purpose of the study is needed. Secondly, I still advise the authors to more clearly indicate what was included in the cultural industries. Information about this is presented in the article only in paragraph 4.1.1 in the caption to the formula. However, the composition of industries requires justification. Thirdly, since the authors left the previous calculations regarding government spending in GDP (general, and not for culture separately). This essential limitation needs to be added to the text. Fourth, I appreciate the authors for adding maps (Figure 2). But I would advise adding a paragraph of text that would contain a description of them. Finelly, it is advisable to move the text about the novelty of the research to the final sections of the article.
